# Peptides Targeting the IF1–ATP Synthase Complex Modulate the Permeability Transition Pore in Cancer HeLa Cells

**DOI:** 10.3390/ijms25094655

**Published:** 2024-04-25

**Authors:** Martina Grandi, Simone Fabbian, Giancarlo Solaini, Alessandra Baracca, Massimo Bellanda, Valentina Giorgio

**Affiliations:** 1Department of Biomedical and Neuromotor Sciences, University of Bologna, 40126 Bologna, Italy; 2Department of Chemical Science, University of Padova, 35121 Padova, Italy; 3Institute of Biomolecular Chemistry of National Research Council of Italy (CNR), 35131 Padova, Italy

**Keywords:** cancer, mitochondria, therapy, ATP synthase, permeability transition pore, inhibitor protein IF1

## Abstract

The mitochondrial protein IF1 is upregulated in many tumors and acts as a pro-oncogenic protein through its interaction with the ATP synthase and the inhibition of apoptosis. We have recently characterized the molecular nature of the IF1–Oligomycin Sensitivity Conferring Protein (OSCP) subunit interaction; however, it remains to be determined whether this interaction could be targeted for novel anti-cancer therapeutic intervention. We generated mitochondria-targeting peptides to displace IF1 from the OSCP interaction. The use of one selective peptide led to displacement of the inhibitor IF1 from ATP synthase, as shown by immunoprecipitation. NMR spectroscopy analysis, aimed at clarifying whether these peptides were able to directly bind to the OSCP protein, identified a second peptide which showed affinity for the N-terminal region of this subunit overlapping the IF1 binding region. In situ treatment with the membrane-permeable derivatives of these peptides in HeLa cells, that are silenced for the IF1 inhibitor protein, showed significant inhibition in mitochondrial permeability transition and no effects on mitochondrial respiration. These peptides mimic the effects of the IF1 inhibitor protein in cancer HeLa cells and confirm that the IF1–OSCP interaction inhibits apoptosis. A third peptide was identified which counteracts the anti-apoptotic role of IF1, showing that OSCP is a promising target for anti-cancer therapies.

## 1. Introduction

The mitochondrial protein IF1 has been characterized as a pro-oncogenic factor playing a role in promoting cancer development and growth [1,2]. This is a 10 kDa heat-stable protein, which is a reversible non-competitive inhibitor of ATP hydrolysis [3,4,5,6,7]. Apart from this physiological role in mitochondria [8,9], the connection of the IF1 protein with the field of cancer started when its high expression level was found associated to different human tumors [10,11,12,13].

In cancer, the inhibitor IF1 seems to play a dual role. On the one hand, it prevents ATP hydrolysis under anoxia/near-anoxia, favoring cell survival and proliferation [14]; on the other, recent findings suggest that IF1 might bind to the ATP synthase complex [15,16,17,18,19] when the enzyme works physiologically [i.e., ATP synthesis [20,21,22,23] by forming an IF1–ATP synthase complex. Accordingly, an alternative IF1 binding site was shown on the ATP synthase oligomycin sensitivity conferring protein (OSCP) in HeLa cells under active oxidative phosphorylation [24]. This study was necessary to clarify the second role of IF1 in cancer, since the interaction of IF1 with the ATP synthase catalytic subunits requires the hydrolysis of two ATP molecules, and it does not occur during ATP synthesis [4,25,26,27,28]. The action of IF1 on this alternative binding site inhibits the permeability transition (PT) pore (PTP) and apoptosis [24].

The mentioned anti-apoptotic role of IF1 occurs through the PTP desensitization to Ca^2+^ ions in the matrix. The PTP is a Ca^2+^-dependent high-conductance channel on the inner mitochondrial membrane [29,30,31], and its prolonged opening leads to apoptosis. Many cancer models inhibit cell death in response to fluctuations of the PTP effectors, such as Ca^2+^ or Mg^2+^, reactive oxygen species (ROS), and matrix pH [32], or controlling the PTP association with its physiological inducer, the cyclophilin D (CyPD) [33,34,35] binding its OSCP subunit [34,36].

The anti-apoptotic effect of IF1 on the OSCP binding site is compatible with other mechanisms that have been associated to the IF1 upregulation in cancer. The IF1 overexpression was shown to keep stable ATP synthase dimers and high cristae density [12,37]. A high IF1 level in cancer might counteract the apoptotic process by OPA1 oligomer stabilization, impeding cristae remodeling during apoptosis [38]. In hepatocellular carcinoma cells, IF1 was shown to promote cell proliferation and colony formation in vitro by decreasing the expression of E-cadherin and increasing the STAT3 level [39].

The identification of peptides targeting the IF1–OSCP subunit interaction, and causing the IF1 displacement from the binding site on the OSCP subunit, which exerts a pro-survival effect in cancer cells, might represent a new strategy for developing anti-cancer therapy.

Moreover, the identification of peptides directed to the OSCP subunit of ATP synthase might be important to confirm the mechanism of binding of IF1 on the novel site and further elucidate the participation of ATP synthase to the permeability transition pore. Several subunits of ATP synthase, although debated [40,41], have been shown to participate in the PTP modulation [42,43,44], including the peripheral stalk subunit OSCP [45,46] and the subunits b and f [47,48]. With this in mind, we generated peptides targeting the IF1–ATP synthase complex and studied their effects on the IF1–OSCP interaction, PTP opening and oxidative phosphorylation in HeLa cancer cells, silenced or not for the pro-oncogenic protein IF1.

## 2. Results

### 2.1. Peptides IF1-O.1 and IF1-O.2 Are Predicted to Target the IF1–OSCP Subunit Interaction

The peptides IF1-O.1 and IF1-O.2 were designed based on the highest sequence homology between the coiled-coil region of IF1 and the N-terminal domain of the OSCP protein [49,50]. IF1-O.1 and its membrane-permeable derivative (TAT-IF1-O.1) target the IF1 sequence, overlapping a small part of the coiled coil domain, which allows the formation of IF1 dimers (Figure 1A) downstream from the residues involved in the IF1–OSCP interaction, as recently found by NMR experiments [24]. IF1-O.2 and its membrane-permeable form (TAT-IF1-O.2) target the N-terminal region of the OSCP subunit, which was previously shown to be involved in the interaction with IF1 (Figure 1B). Moreover, a third peptide IF1-O.3 and its membrane-permeable form (TAT-IF1-O.3) were generated in order to target the C-terminus of the OSCP subunit, which was not targeted by IF1 binding but is very important to stabilize the interactions of the OSCP with the other peripheral stalk subunits in the assembled ATP synthase [51]. 

The primary sequences of the aminoacidic regions that are predicted to be targeted by the three peptides are shown in Figure 1C.

### 2.2. Peptide IF1-O.1 Displaces IF1 from ATP Synthase and IF1-O.2 Interacts with the IF1 Binding Site on the OSCP Subunit

In order to assess the effects of the peptides in targeting the IF1–OSCP interaction in isolated mitochondria from HeLa cells, the whole ATP synthase enzyme was immunoprecipitated after extraction in its native form. The ATP synthase extract was prepared with the addition of 1% digitonin (*w*/*v*) in an aminocaproic acid-based buffer containing mitochondria that had been kept under active oxidative phosphorylation in an ADP-regenerating buffer. This condition was previously shown to promote the IF1 binding to the OSCP subunit [24] in different cancer models. The treatment with the IF1-O.1 peptide during oxidative phosphorylation, followed by mitochondrial extraction and immunoprecipitation of ATP synthase, caused a decrease in the IF1 level which was immunoprecipitated with the enzyme (Figure 2A). This effect on the IF1 binding to the enzyme was not observed when the IF1-O.2 or IF1-O.3 peptides were used in incubation (Figure 2A), confirming the specificity of IF1-O.1 for targeting the IF1 inhibitor and causing its displacement from the OSCP subunit under ATP synthesis condition. The absence of a displacing effect upon IF1-O.2 treatment might be caused by the binding of IF1 to the OSCP subunit, which makes the OSCP subunit inaccessible to this peptide in wild-type HeLa mitochondria. The interaction of IF1-O.2 with the OSCP subunit was tested in vitro through NMR experiments by using the recombinant N-terminal domain (R6-G114) of the human OSCP subunit of ATP synthase, which was previously found to be involved in the interaction with IF1 [24]. The comparison of ^1^H-^15^N SOFAST-HMQC spectra of the N-terminus of OSCP in the presence or absence of the IF1-O.2 peptide (in a 10-fold molar excess) indicated that a number of peaks were shifted upon the addition of this peptide (Figure 2B). On the contrary, when the IF1-O.3 peptide was added, no signal perturbations were detected (Appendix A), which is in line with the prediction that this peptide binds to the C-terminus of OSCP (Figure 1B,C). The peaks that were shifted upon the IF1-O.2–OSCP interaction are located mostly in the helices 1 and 5 of the OSCP subunit (Figure 3A, upper panel), overlapping two of the three regions of amino acids that are shifted upon the IF1–OSCP interaction, which were obtained with the incubation of the N-terminus of OSCP with the G1-E40 N-terminal fragment of IF1 (Figure 3A, lower panel). 

To verify if these regions are accessible for the interaction during the ATP synthase catalytic activity, we considered the N-terminus of OSCP assembled with the rest of ATP synthase, using the human cryo-EM structure of the enzyme (PDB code 8H9V, [51]). This analysis showed that the residues in helix 1 (Figure 3A,B, blue) and helix 5 (Figure 3A,B, red) form a continuous area exposed on the surface of the protein (Figure 3C) which is available for the interaction with IF1 or IF1-O.2 peptide and is more exposed when the enzyme is in state 3 conformation of the rotary catalysis (Figure 3C(i,iii)) compared to state 1 (Figure 3C(ii,iv)). Each rotary state is determined by the position of the rotor relative to the stator during the catalytic activity, as shown in [51].

These results lead us to the conclusion that the interaction between IF1 and the N-terminal domain of OSCP can be targeted by mitochondria-directed peptides. Two peptides showing sequence homology with specific fragments of the IF1 or OSCP proteins were able (i) to displace IF1 from the interaction during oxidative phosphorylation and (ii) mimic the molecular shift perturbation on the N-terminus of OSCP caused by IF1, matching the IF1–OSCP interaction. 

These results support that the regions of the OSCP subunit that are involved in the IF1–OSCP interaction, as modeled on the human ATP synthase cryo-EM structure, are favored when the enzyme is in state 3 conformation during its catalytic activity. 

### 2.3. Peptides IF1-O.1, IF1-O.2 and IF1-O.3 Do Not Affect Mitochondrial Bioenergetics

Knocking down IF1 (IF1 KD) adherent HeLa cells in which the IF1 levels were decreased by about 80–85% (Figure 4A) did not show different mitochondrial respiration than relative controls under basal conditions upon the addition of the ATP synthase inhibitor oligomycin, the complex I inhibitor rotenone or the complex III inhibitor antimycin A (Figure 4B and Appendix A). A mild but not significant decrease in oxygen consumption was revealed following the effect of the uncoupler in control cells (FCCP, Figure 4B). In line with the comparable respiratory profile of IF1 KD and control HeLa cells, subunits that are important in OXPHOS assembly were observed to be unchanged in cell lysates as shown by the quantification of their bands (Figure 4C). To assess the effects of mitochondria-targeting peptides on mitochondrial function, the membrane-permeable derivatives of IF1-O.1, IF1-O.2, and IF1-O.3 were used to treat the cells in situ. IF1 KD and control HeLa cells were used in order to figure out any effect of peptides that might be highlighted in the absence or presence of the IF1 protein. The treatment with TAT-IF1-O.1 (Figure 5A), TAT-IF1-O.2 (Figure 5B) or TAT-IF1-O.3 (Figure 5C) peptides up to the concentration of 30 µM did not show any significant effect on cellular respiration either in control (i) or IF1 KD (ii) HeLa cells, as shown by their normalization for the cellular protein (Appendix A). Moreover, the peptide treatment did not affect the mitochondrial membrane potential of HeLa cells, nor the OXPHOS maximal activity (as shown by the comparison between the basal and FCCP-stimulated respiration, upon peptide treatment, Figure 5). These results indicate that these peptides do not cause in the short-term treatment of any cellular toxicity and confirm that their binding to the IF1–ATP synthase complex does not modulate oxidative phosphorylation.

### 2.4. Peptides IF1-O.1, IF1-O.2 and IF1-O.3 Modulate the Permeability Transition Pore Opening

The analysis of the role of IF1 in the modulation of the PTP showed that the Ca^2+^ threshold which promotes the PTP opening was significantly more decreased in IF1 KD than in control cells, and it was kept in State 3 respiration (Figure 6A). This indicated that IF1 desensitizes respiring cancer HeLa cells to PTP opening in a mechanism which seems independent of CyPD [24]. We further tested the role of TAT-IF1-O.1 (Figure 6B), TAT-IF1-O.2 (Figure 6C) or TAT-IF1-O.3 (Figure 6D) on PTP modulation in permeabilized HeLa cells that were grown in a high-glucose-containing medium, and these were subjected to continuous oxidative phosphorylation to promote the IF1 binding to the OSCP subunit (Figure 6).

The Ca^2+^ concentration activating PT was significantly higher in IF1 KD HeLa cells that were treated with either TAT-IF1-O.1 (Figure 6B) or TAT-IF1-O.2 (Figure 6C) peptides compared to the untreated counterpart. These results indicate that both peptides mimic the effect of the native IF1, in inhibiting the PTP opening, in IF1 KD cells. On the other hand, the TAT-IF1-O.1 (Figure 6B) and TAT-IF1-O.2 (Figure 6C) peptides did not show any effect in control HeLa cells, suggesting that the high expression level of IF1 in this cellular model causes a dynamic of reassociation of the native protein in intact mitochondria, after its displacement upon peptide IF1-O.1 treatment. However, this interpretation is not in contrast with the IF1 displacement (Figure 2A) observed through immunoprecipitation experiments. In fact, the use of digitonin during immunoprecipitation causes the release of IF1 from the matrix and may avoid its re-binding to the OSCP subunit.

Very interestingly, the TAT-IF1-O.3 which targets the C-terminus of the OSCP subunit, as predicted in Figure 1B,C, affects PTP opening with a biphasic behavior in both IF1 KD and control HeLa cells (Figure 6D). The PTP opening is sensitized by this peptide at a lower concentration in IF1 KD cells (30 µM) than their controls (50 µM), (Figure 6D). However, a 48 h treatment with 30 µM TAT-IF1-O.3 did not show any toxicity on proliferation in both cell lines (Appendix A). This result suggests that the latter peptide can promote PTP opening (without any toxic effect) independently of the IF1–OSCP interaction in control cells.

## 3. Discussion

This study shows that the mitochondria-targeting peptide IF1-O.1 is able to disrupt the IF1–OSCP interaction. The interaction of IF1-O.2 with the OSCP subunit studied by NMR experiments simulated the IF1 binding to the OSCP subunit in regions that are exposed on the enzyme surface during the catalytic activity of ATP synthase. Importantly, the use of mitochondria-targeting peptides directed to the IF1–OSCP interaction or to the C-terminal region of the OSCP subunit mimic or counteract, respectively, the effects of the high levels of IF1 in HeLa cancer cells by modulating the PTP opening.

IF1 is overexpressed in many human tumors [1,13,32], and different hypotheses on its pro-oncogenic role have been proposed [14,32,38,52]. IF1 increases with the tumor progression of NSCLC biopsies [53] and was found to be upregulated in lung tumors compared to control tissues [13]. Its expression is significantly increased in bladder cancer, glioma and hepatocarcinoma biopsies [54,55,56]. Moreover, in mouse xenografts, the IF1 silencing showed decreased tumor growth and metastasis [56], pointing out the role of this inhibitor in vivo. Finally, the injection of IF1 KO or control HeLa cells in zebrafish embryos recapitulated the same effects of IF1 [24] on tumor mass development and metastasis shown in mice. Although the role of this protein in promoting more aggressive phenotypes was not confirmed in breast and colorectal tumors [1], the overexpression in non-tumoral MEF cells of one mutant form of IF1, which can only bind on the OSCP subunit, was sufficient to inhibit the PTP opening and induce a tumorigenic phenotype [24].

MEFs overexpressing IF1 showed that colony formation in soft agar depends on the IF1 expression level. This result is in line with previous findings showing that IF1 KD decreased colony formation and migration in hepatocellular carcinoma cells [39], and it is consistent with the limited colony formation previously reported in IF1 KD and IF1 KO HeLa models [24,38]. In the aforementioned models, the IF1 upregulation controls the colony-forming capacity through the inhibition of the PTP-dependent apoptosis, without affecting cellular respiration or proliferation [24], as confirmed by the comparison of control and IF1 KD HeLa cell oxygen consumption and calcium retention capacity analyses, as shown in this work. Indeed, the treatment of the IF1 KD HeLa cells with the peptides IF1-O.1 and IF1-O.2, targeting two regions in IF1 and in the N-terminal part of the OSCP subunit, respectively, caused a dose-dependent inhibition of the PTP opening, confirming that this event is controlled by the IF1–OSCP or peptide–OSCP interaction. In line with this interpretation, these peptides did not show their effects in cells harboring high levels of the endogenous IF1, re-binding or masking the OSCP binding site.

The IF1–OSCP subunit interaction appears crucial in the anti-apoptotic action of IF1 in cancer cells. Previous NMR studies showed that this site on the OSCP subunit is bound by IF1 with lower affinity than the canonical site in the catalytic core on ATP synthase [24], which is in line with the evidence that this interaction occurs in a cancer model in which IF1 is upregulated. The IF1–OSCP interaction occurs under active oxidative phosphorylation, which is a condition in which the canonical binding site is not available for IF1 [28,57]. The use of the IF1-O.2 peptide in NMR studies showed that the residues of the OSCP subunit that are shifted upon the peptide binding are localized in two of the three regions that were perturbated by the binding of IF1 [24]. These regions include parts of helices 1 and 5, allowing a further characterization of the site of interaction on the OSCP subunit. Moreover, the fact that helices 1 and 5 were subjected to the same perturbations upon treatment with IF1-O.2 suggests that the IF1–OSCP interaction can be pharmacologically targeted. Meanwhile, the effect of IF1-O.1 on the immunoprecipitation shows that IF1 can be displaced by ad hoc peptides.

Our results also indicate that the residues involved in the IF1– or peptide–OSCP subunit interaction are exposed on the ATP synthase surface during the catalytic activity, supporting the fact that this novel binding site is accessible under oxidative phosphorylation [24]. 

Although the role of the OSCP subunit in the modulation of PT has been proposed [34,35,46,58,59], the peptides that target the IF1–OSCP interaction used in this paper were not effective in promoting PTP opening in IF1 expressing HeLa cells, as it was instead described for benzodiazepine 423, which displaces IF1-sensitizing cells to PTP-dependent apoptosis [24]. 

Importantly, a third peptide which is directed on the C-terminus of the OSCP subunit was assayed in the HeLa cancer cell model and succeeded in the activation of the PTP. The region of the OSCP subunit which is targeted by this peptide has an important role in the coupling of ATP synthase by connecting the top of the catalytic subunits with the peripheral stalk of the enzyme [51,60]. Moreover, in a bioinformatic study aimed at identifying the molecular rearrangements occurring on the OSCP subunit during the Ca^2+^ binding to the ATP synthase and allowing PTP opening, the C-terminus of OSCP was predicted to be subjected to fluctuations [46]. This fact is compatible with the effect of peptide IF1-O.3 targeting the C-terminal region of the OSCP subunit and promoting PTP opening. This latter result indicates the protein fragment which can be targeted in future studies to develop anti-cancer compounds. In order to better define the therapeutic potential of the synthetic peptides shown here, that were useful to address the molecular mechanism of the IF1–OSCP binding in HeLa cells, their effects should be tested on other cell types and in in vivo cancer models.

## 4. Materials and Methods

### 4.1. Stable Knocking down of IF1 Protein

Stable interference of IF1 protein (IF1 KD) was obtained by HeLa transfection with a shRNA (sequence: 5′-GATATTTCCGAGCACAGAGTA-3′) cloned into the pLKO.1 lentiviral vector and targeting the human IF1 mRNA. ATP5IF1 shRNA and its corresponding control (CTR) vector, pLKO.1, were purchased from Sigma (SHC001 Sigma, St. Louis, MO, USA). This shRNA plasmid was co-transfected with the packaging plasmids VSV-G and p8.74 into HEK 293 T cells for viral production. Recombinant virus was collected and used to infect HeLa cells by standard methods. Cells were selected and maintained, in 0.8 μg/mL puromycin for a stable knocking down, and analyzed by Western blotting. The selected mixed cell populations did not follow the single-cell cloning procedure, but IF1 KD cells maintained less than 20% of residual IF1 compared to controls.

### 4.2. Experimental Model

Wild-type, plko and IF1 KD HeLa cells were cultured in Dulbecco’s modified Eagle’s medium (DMEM, Thermo Fisher Scientific, Waltham, MA, USA), supplemented with fetal bovine serum (FBS, 10% *v*/*v*, Thermo Fisher Scientific), glutamine (4 mM, Thermo Fisher Scientific), penicillin and streptomycin (1% *v*/*v*, Thermo Fisher Scientific) at 37 °C with 5% CO_2_. The media for plko and IF1 KD HeLa cells were supplemented with 0.8 μg/mL puromycin to maintain the selection.

### 4.3. Peptide Synthesis

A rational peptide design approach, as previously published [61], has been followed to find a specific inhibitor of the IF1–OSCP interaction. Human IF1 (Q9UII2; UniProtKB) and OSCP (ATP synthase subunit O, P48047; UniProtKB) protein sequences were aligned by Pearson’s lalign program on the ExPASy server. Aligned sequences, and corresponding linked TAT (derived from TAT_47–57_) cell-permeable peptides were synthesized by Ontores Biotechnologies, Hangzhou, China. The purity of peptides was >90% measured by RP-HPLC Chromatogram.

### 4.4. Immunoprecipitation of ATP Synthase

Immunoprecipitation of ATP synthase was performed in wild-type HeLa cells. Freshly prepared mitochondria were obtained by using a glass–Teflon potter as previously described [62]. The mitochondria (500 µg) were resuspended in 500 µL buffer promoting state 3 respiration (0.1 M glucose, 80 mM KCl, 10 mM Mops-Tris, 5 mM succinate-Tris, 4 mM MgCl_2_, 1 mM Pi-Tris, 0.5 mM NADP^+^ 0.4 mM ADP, 50 μM P1, P5-di (adenosine-5′) pentaphosphate, 10 μM EGTA, 4 U/mL glucose-6-phosphate dehydrogenase, and 3 U/mL hexokinase, pH 7.4) and incubated with or without 30 µM membrane-permeable peptides for 10 min at room temperature. Mitochondria were then centrifuged at 8000× *g* for 10 min at 4 °C, resuspended in 50 µL of buffer containing 0.75 M aminocaproic acid, 50 mM Bis-Tris, pH 7.0, and solubilized with digitonin (1% *w*/*v*, Sigma-Aldrich, St. Louis, MO, USA). After centrifugation at 100,000× *g* for 25 min at 4 °C, the supernatant was collected. Extracted proteins were supplemented with 10 μL of anti-complex V monoclonal antibody covalently linked to protein G-sepharose beads (ATP synthase immunocapture kit, Abcam, Cambridge, UK), and incubated overnight at 4 °C under wheel rotation. For the elution, beads were incubated three times with sample buffer 1× for 15 min, and the collected fractions (eluates) were subjected to SDS-PAGE, which was followed by Western blotting for β subunit and IF1 [24,34].

### 4.5. Nuclear Magnetic Resonance

^15^N-labeled OSCP-NT for the NMR experiments was produced as previously reported [24]. Briefly, after protein expression in *E. coli* BL21 (DE3) using 1g/L ^15^NH_4_Cl as a nitrogen source, OSCP-NT was purified by three chromatographic steps with metal ion affinity chromatography (Ni^2+^-HisTrapTM FF column, GE Healthcare, Chicago, IL, USA), cation exchange chromatography (HiTrap^TM^ SP HP column, GE Healthcare) and size-exclusion chromatography (Superdex 75 PG HiLoad 16/60 column, GE Healthcare) equilibrated in 50 mM sodium phosphate, 300 mM NaCl, pH = 7. The protein was finally concentrated to 300 μM (Vivaspin 5kDa MWCO Merck, Darmstadt, Germany, 5500 × g × 15 min, 4 °C) for subsequent interaction experiments. 

All NMR experiments were performed with a Bruker AVANCE NEO 600 MHz spectrometer equipped with a 5 mm cryogenic probe Prodigy TCI. All NMR spectra were processed with Bruker Topspin 4.0.6 (Bruker BioSpin GmbH, Rheinstetten, Germany) and analyzed using NMRFAM-Sparky [63]. The SOFAST-HMQC experiments [64] were acquired with 128 scans, a recovery delay of 150 ms before each scan, and 160 increments in the indirect dimension.

To study the interaction between OSCP-NT and IF1-O.2 or IF1-O.3 peptides by NMR, ^1^H-^15^N SOFAST–HMQC spectra were acquired on ^15^N-labeled OSCP-NT (final concentration 70 μM) with and without 10-fold molar excess of the unlabeled peptide at 25 °C (all peptides were in Milli-Q ultrapure water, stock solutions of 1mM). Around 290 μL of each sample was loaded into a Shigemi microtube (Shigemi Inc., Allison Park, PA, USA) for data collection. The final buffer composition in each NMR tube was 20 mM sodium phosphate, 80 mM NaCl, D_2_O 4% (*v*/*v*), pH = 7.

Chemical shift perturbation (Δ*δ*) for each OSCP-NT cross-peak was evaluated using the following equation: $$∆\delta =\sqrt{\frac{{\left(∆\delta H\right)}^{2}+{\left(\frac{∆\delta N}{5}\right)}^{2}}{2}}$$
where Δ*δH* and Δ*δN* represent the ^1^H and ^15^N amide chemical shift changes, respectively. The standard deviation σ, calculated across the entire sequence, was used as the threshold value: residues with Δ*δ* > σ were considered significantly perturbed. The significantly perturbed residues were mapped on the isolated OSCP-NT by using the human domain structure (6–114) obtained by the cryo-EM structure of the Homo Sapiens ATP synthase in state 3b (PDB code 8H9V) and also on the whole OSCP subunit (1–190) anchored to the rest of the enzyme, using the cryo-EM structure of the *Homo sapiens* ATP synthase in state 1 (PDB code 8H9S) and in state 3b. All structures were represented using Pymol (version 1.8.4 Open Source).

### 4.6. Oxygen Consumption Rate

The oxygen consumption rate (OCR) in adherent cells was measured using the XF24 Extracellular Flux Analyzer (Agilent technologies, Santa Clara, CA, USA). Plko and IF1 KD HeLa cells were seeded in XF24 cell culture microplates at 50,000 cells/well and left to grow at 37 °C in a 5% CO_2_ humidified incubator for 24 h. The day after, the growth medium was replaced with the Seahorse medium (DMEM, Sigma D5030, supplemented with NaCl, glutamine and phenol red according to the manufacturer protocol and 25 mM glucose, 10 mM sodium pyruvate), and cells were incubated at 37 °C for 30 min to allow temperature and pH equilibration. During this period, cells were treated with or without 30 µM membrane-permeable peptides. After an OCR baseline measurement, 1 µM oligomycin, 0.2 µM FCCP, 1 µM rotenone, and 1 µM antimycin A were sequentially added to each well. Before each experiment, a titration with FCCP was performed to determine the optimal FCCP concentration that stimulates respiration maximally.

### 4.7. Calcium Retention Capacity

For the Calcium Retention Capacity (CRC) assay, external mitochondrial Ca^2+^ was measured by Ca^2+^ Green-5N fluorescence using a Tecan Infinite^®^ 200 PRO (Tecan Trading AG, Männedorf, Switzerland) plate reader. Plko and IF1 KD HeLa cells were permeabilized as mentioned in [48] and resuspended at the concentration of 10^7^ × ml^−1^ in a buffer promoting state 3 respiration (0.1 M glucose, 80 mM KCl, 10 mM Mops-Tris, 5 mM succinate-Tris, 4 mM MgCl_2_, 1 mM Pi-Tris, 0.5 mM NADP^+^, 0.4 mM ADP, 50 μM P1, P5-di (adenosine-5′) pentaphosphate, 10 μM EGTA, 4 U/mL glucose-6-phosphate dehydrogenase, and 3 U/mL hexokinase, pH 7.4) and treated with or without 10–50 µM membrane-permeable peptides. For all CRC measurements, sequential 2.5 or 5 µM CaCl_2_ pulses were added to cells and Ca^2+^ Green-5N fluorescence was measured.

### 4.8. Lysates, Gel Electrophoresis, Western Blotting

Cells (10 × 10^6^) were kept on ice for 20 min in 0.15 mL of a buffer containing 150 mM NaCl, 20 mM Tris, 5 mM EDTA-Tris, pH 7.4 with the addition of Triton X-100 (1% *v*/*v*, Sigma-Aldrich) and glycerol (10% *v*/*v*, Sigma-Aldrich). Sample buffer (NuPAGE™ LDS Sample buffer, Invitrogen, Waltham, MA, USA) supplemented with β-mercaptoethanol (12.5% *v*/*v*, Sigma-Aldrich) was added to supernatants, and samples were separated by polyacrylamide gel (NuPAGE™, 12% Bis-Tris, Invitrogen) electrophoresis and transferred to nitrocellulose membranes. Blocking was performed with a PBS solution containing 5% (*w*/*v*) non-fat dry milk (AppliChem, Darmstadt, Germany). Antibodies for inhibitor factor 1 (IF1), β subunit and OXPHOS (Human WB Antibody Cocktail) were from Abcam (Cambridge, UK). Band pixels of each replicate are normalized on band pixels of their proper loading control. Mean pixel ratios ± SEM are then shown.

### 4.9. Growth Curve

HeLa cells (plko and IF1 KD) were seeded at 7500 cells/well in a 96-well tissue culture plate in DMEM containing 25 mM glucose in the presence or absence of 30 µM TAT-IF1-O.3 peptide. Cells were incubated at 37 °C in a 5% CO_2_ humidified incubator and cultured for 48 h. After 24 and 48 h, cells were harvested with trypsin and counted. 

### 4.10. Quantification and Statistical Analysis

Unless otherwise stated in the figure legends, each experiment derives from at least three independent biological replicates. Data are presented as mean ±  SEM. *p* values indicated in the figures are calculated with GraphPad (version 8.0.1), two-way ANOVA or Student’s *t* test are applied (* represents *p*  ≤  0.05, ** *p*  ≤  0.01, *** *p*  ≤  0.001). The variance between the groups that are compared is similar. Western blotting band intensities were analyzed using a ChemiDoc MP system equipped with the ImageLab software (version 6.1, Bio-Rad, Hercules, CA, USA) or ImageJ software (version 1.52p).

## 5. Conclusions

In conclusion, this study elucidates the effects of three peptides targeting the IF1–ATP synthase complex. Two peptides directed to the IF1–OSCP interaction mimic the effects of the IF1 binding to its site of interaction, inhibiting PTP opening in HeLa cells that are deprived of IF1. These peptides might be considered for future studies in ischemia or neuromuscular disease models, in which the inhibition of PTP-dependent apoptosis can be considered as a therapeutic strategy. A third peptide, which on the contrary promotes PTP opening, discloses an OSCP C-terminal aminoacidic region as a promising target to develop new anti-cancer therapies. 

## Figures and Tables

**Figure 1 ijms-25-04655-f001:**
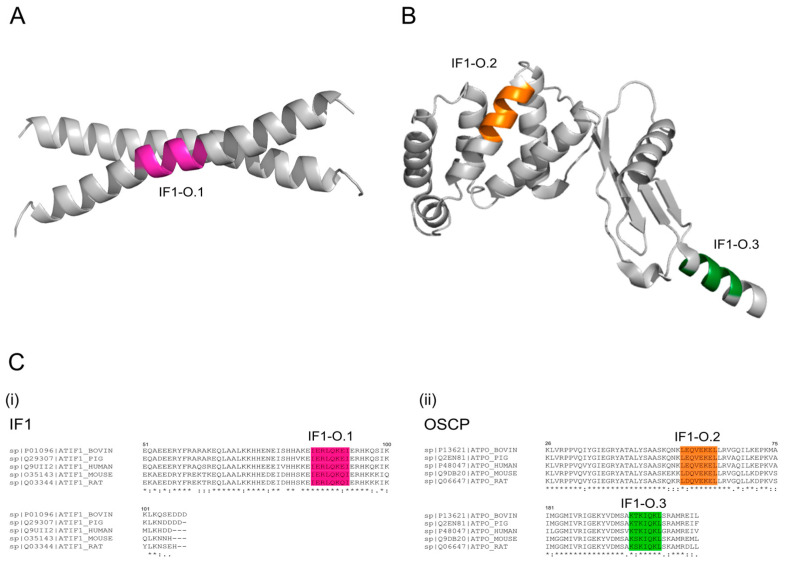
Amino acid sequences of mitochondria-targeting peptides and predicted binding sites on IF1 or OSCP proteins. In (**A**), a cartoon representation of the C-terminal coiled-coil domain from bovine IF1 (PDB code 1HF9) showing the region targeted by peptide IF1-O.1, colored in magenta. In (**B**), a cartoon representation of the OSCP subunit from the cryo-EM structure of *Homo Sapiens* ATP synthase (PDB code 8H9V) in state 3 conformation of the rotary catalysis showing the regions targeted by the peptide IF1-O.2 and peptide IF1-O.3, colored in orange and green, respectively. In (**C**), multiple alignments are shown of the human amino acid sequences (harboring the MTS) for (**i**) IF1 (residues 51–106) and (**ii**) OSCP subunit (residues 26–75 and 181–213) with their equivalent sequences from ox, pig, mouse and rat. The predicted sequences that are targeted by peptide IF1-O.1, peptide IF1-O.2 and peptide IF1-O.3 are shown in magenta, orange and green, respectively. The UniProt accession numbers for proteins from *Bos taurus*, *Sus scrofa*, *Homo sapiens*, *Mus musculus*, *Rattus norvegicus* are shown on the left of each sequence. An asterisk (*) indicates positions which have a single, fully conserved residue; a colon (:) indicates conservation between amino acid groups or similar properties.

**Figure 2 ijms-25-04655-f002:**
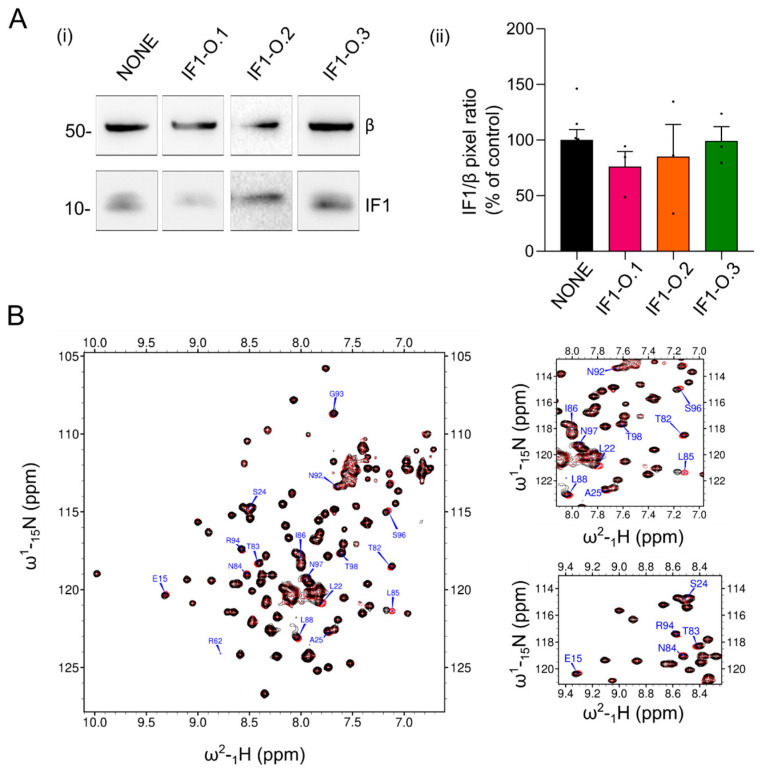
Peptide IF1-O.1 displaces IF1 from ATP synthase and IF1-O.2 interacts with the IF1 binding site on the OSCP subunit. In (**A**), mitochondria isolated from HeLa wild-type cells are incubated in a buffer promoting state 3 respiration in the presence or absence of 30 μM TAT-IF1-O.1, TAT-IF1-O.2 and TAT-IF1-O.3 peptides and are solubilized in a 1% (*w*/*v*) digitonin-containing buffer. Mitochondrial extracts are subjected to immunoprecipitation of ATP synthase. Western blotting shows (**i**) β subunit and IF1 protein in the immunoprecipitated fractions. In (**ii**), the mean of IF1/β pixel ratio is shown (expressed as % of control). Data are mean ± SEM of three independent experiments. In (**B**), ^1^H-^15^N SOFAST HMQC spectra of OSCP-NT (residues R6-G114) in the absence (red) and in presence (black) of a 10-fold molar excess of peptide IF1-O.2. The expanded plots show a region of the spectrum containing some of the residues with a pronounced chemical shift perturbation, which are labeled.

**Figure 3 ijms-25-04655-f003:**
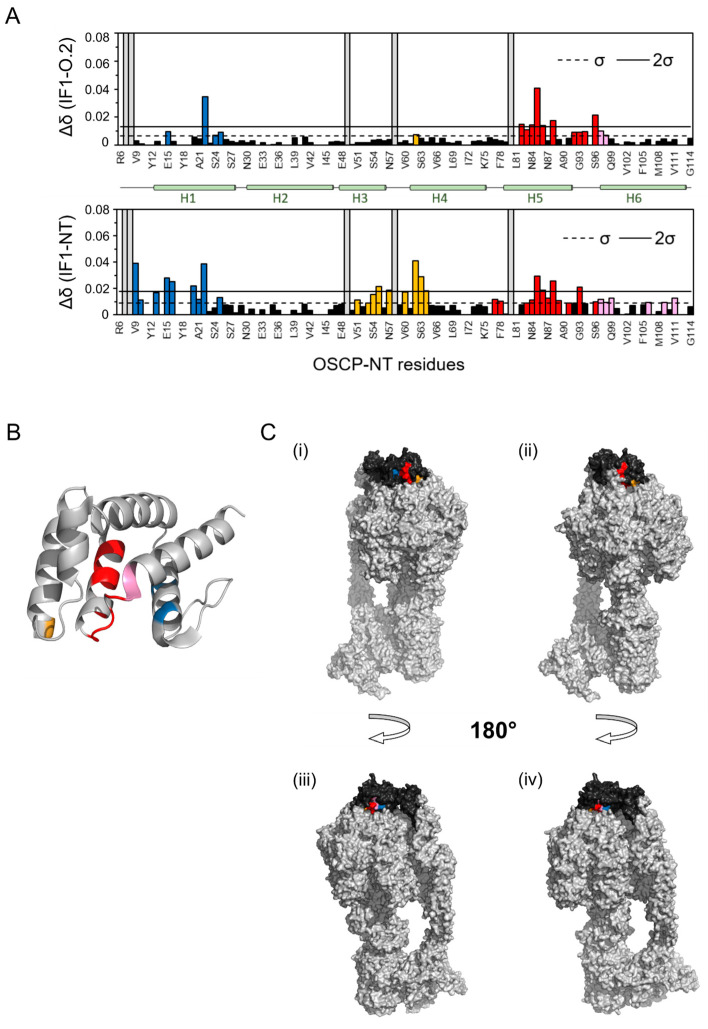
The OSCP amino acidic regions that are shifted upon the IF1-O.2 peptide- or IF1–OSCP interaction and their localization on the human ATP synthase. In (**A**), comparison between chemical shift perturbation (Δ*δ*) calculated from ^1^H-^15^N SOFAST HMQC spectra of OSCP-NT (residues R6-G114) in the absence or presence of 10-fold molar excess of peptide IF1-O.2 or 20-fold molar excess of unlabeled IF1-NT (residues G1-E40). Blue, yellow, red and pink bars indicate residues that belong, respectively, to helix 1 (H1), helix 4 (H4), both helix 5 and the loop region between helices 5 and 6 (H5–H6), and helix 6 (H6), showing a deviation larger than one standard deviation (σ, dashed line). Residues with Δ*δ* larger than two standard deviations are above the continuous line. In (**B**), a schematic representation of OSCP-NT secondary structures is reported. In (**C**), a surface representation of OSCP-NT from the cryo-EM structure of *Homo Sapiens* ATP synthase in state 3 (PDB code 8H9V), (**i**,**iii**) and in state 1 (PDB code 8H9S), (**ii**,**iv**) each showing two different orientations. The OSCP subunit is colored in black. Residues of OSCP-NT affected by the interaction with IF1-O.2 (with Δ*δ* > σ) are colored as described above. Only residues accessible on the surface of the protein are visible.

**Figure 4 ijms-25-04655-f004:**
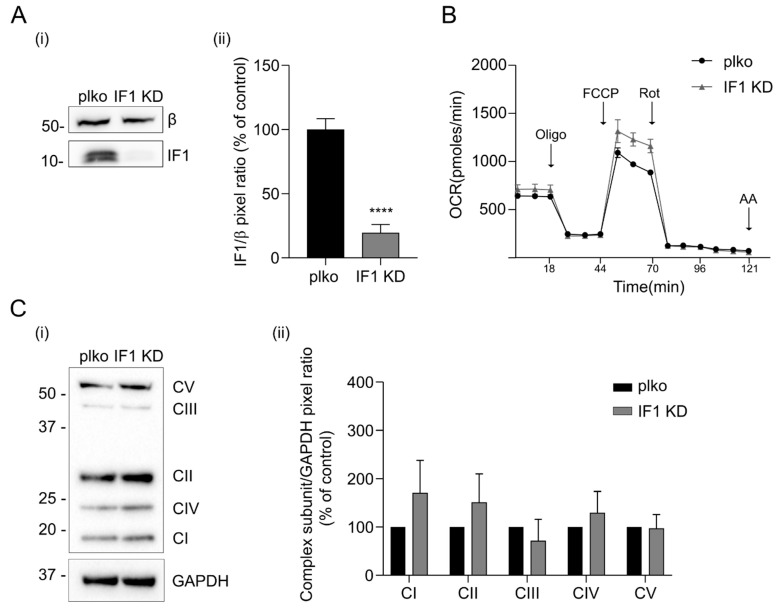
Knocking down the IF1 inhibitor protein does not alter respiration nor OXPHOS complex levels in mitochondria. In (**A**), Western blotting shows β subunit and IF1 protein levels in both control (plko) and IF1 KD HeLa cell lysates (**i**) and the mean of the IF1/β subunit pixel ratio (**ii**), (expressed as % of control). Data are mean ± SEM of four independent experiments. *p* value is **** *p* < 0.0001; Student’s *t* test. In (**B**), representative oxygen consumption rate (OCR) traces of adherent control and IF1 KD HeLa cells in situ. OCR is measured before (basal) and after treatment with oligomycin (oligo), carbonyl cyanide p-(trifluoromethoxy) phenylhydrazone (FCCP), rotenone (Rot) and antimycin A (AA). In (**C**), Western blotting (**i**) is shown of the indicated OXPHOS complexes, GAPDH (glyceraldehyde dehydrogenase as loading control) in control and IF1 KD HeLa cells. The molecular marker is indicated on the left. In (**ii**), the mean ratio is analyzed of band pixels between each complex subunit and GAPDH (mean ± SEM of four independent experiments).

**Figure 5 ijms-25-04655-f005:**
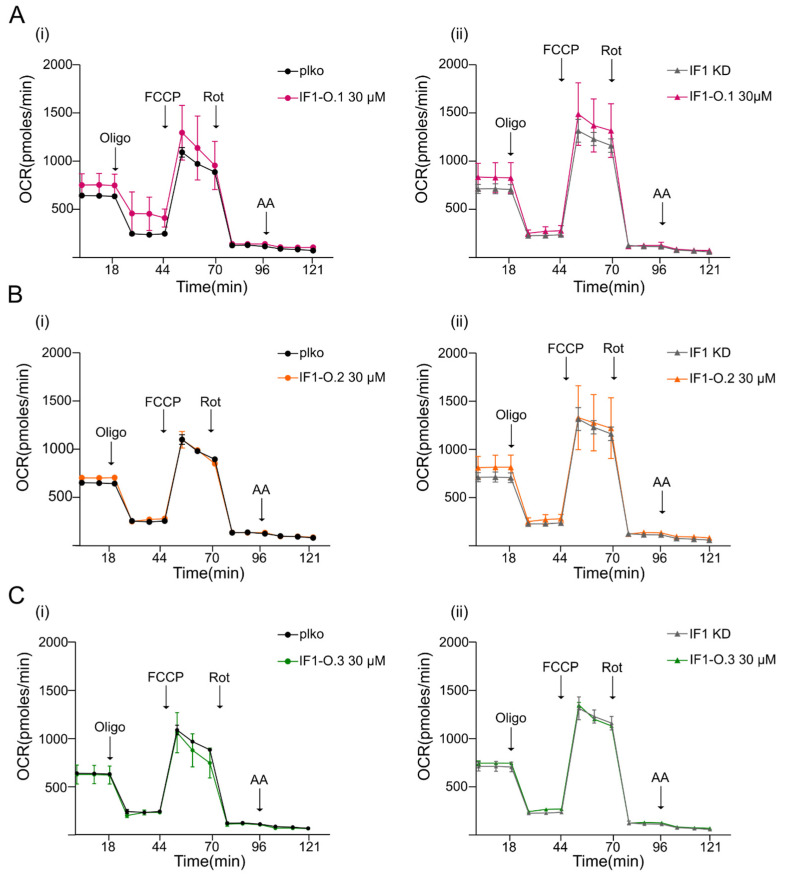
Effects of peptides IF1-O.1, IF1-O.2 and IF1-O.3 on mitochondrial respiration. Traces are shown of the oxygen consumption rate (OCR) by adherent plko (**i**) and IF1 KD (**ii**) HeLa cells, treated with or without 30 μM of membrane-permeable peptides. OCR is measured before (basal) and after treatment with oligomycin (oligo), carbonyl cyanide p-(trifluoromethoxy) phenylhydrazone (FCCP), rotenone (Rot) and antimycin A (AA). In (**A**), TAT-IF1-O.1; in (**B**), TAT-IF1-O.2 and in (**C**), TAT-IF1-O.3 are used for a 30 min treatment before measurements. Data are mean ± SEM of three independent experiments.

**Figure 6 ijms-25-04655-f006:**
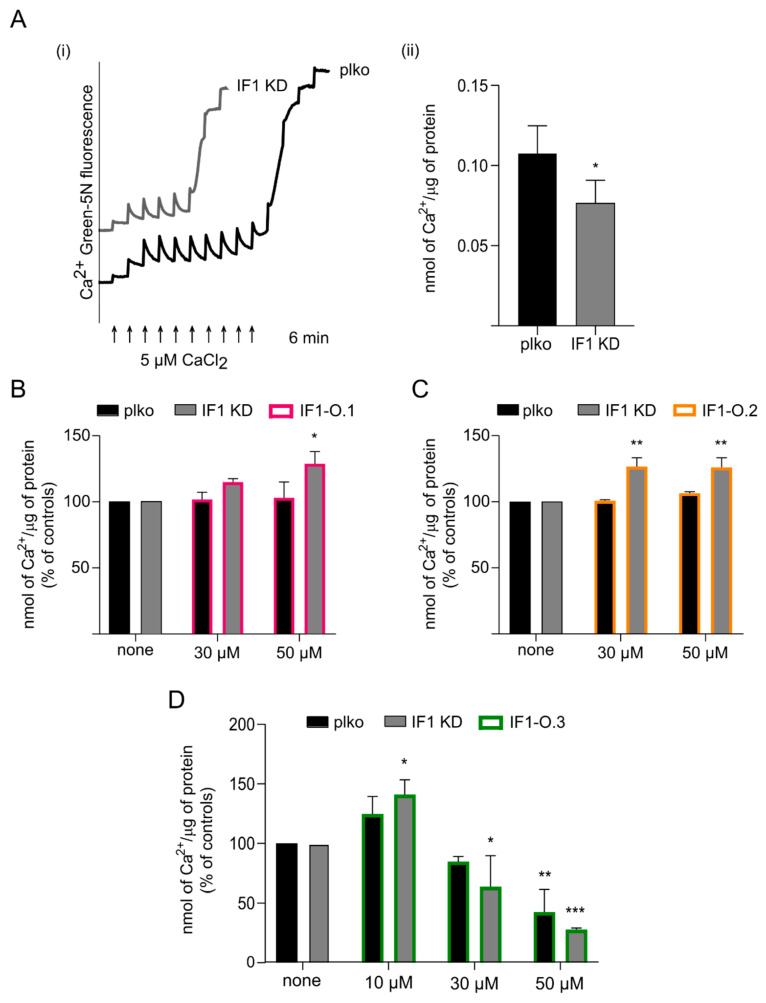
Effects of peptides IF1-O.1, IF1-O.2 and IF1-O.3 on mitochondrial permeability transition pore opening. Ca^2+^ retention capacity (CRC) is assessed in permeabilized control (plko) and IF1 KD HeLa cells in an ADP-regenerating buffer containing respiratory substrates and the membrane-impermeable Ca^2+^ sensor, Ca^2+^ Green-5N. Ca^2+^ Green-5N fluorescence was monitored following repeated additions of 5 μM Ca^2+^ pulses (bottom arrows). A return of Ca^2+^ Green-5N fluorescence to baseline reflects an uptake of Ca^2+^ by mitochondria, whereas a sudden increase in fluorescence is indicative of PTP opening. In ((**A**(**i**)), one representative experiment is shown out of 4 for plko and IF1 KD HeLa cells. Ca^2+^ Green-5N fluorescence baselines are graphically shifted upward to avoid trace overlap. The histogram (**A**(**ii**)) represents nmols of Ca^2+^ per μg of protein retained by plko and IF1 KD cells. Data represent the mean ± SEM (4 independent experiments), the *p* value is * *p* = 0.018, Student’s *t* test. In (**B**–**D**), CRC is assessed as above in plko and IF1 KD permeabilized HeLa cells in the presence of 0–50 μM of the indicated membrane-permeable peptides (TAT-IF1-O.1; TAT-IF1-O.2 and TAT-IF1-O.3). Data are mean (expressed as % of controls) ± SEM of 3 independent experiments. *p* values are * *p* ≤ 0.05, ** *p* ≤ 0.01, *** *p* = 0.0001; two-way ANOVA.

## Data Availability

The datasets generated by this study will be made available by the corresponding authors upon request.

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
