# Peer review of "Peptides Targeting the IF1–ATP Synthase Complex Modulate the Permeability Transition Pore in Cancer HeLa Cells"

_ijms, 2024, doi:10.3390/ijms25094655_

Round 1
Reviewer 1 Report
Comments and Suggestions for Authors
The manuscript entitled: "Peptides targeting the IF1-ATP synthase complex modulate the permeability transition pore in cancer cells" was revised. The manuscript was presented in order in all sections. The design was optimal and adequate to answer the permeability transition pore question. The manuscript has minor observations:
1. The title must be more adequate. According to their results, it must be "in HeLa cells" or "tumoral cell line HeLa" because authors can not associate it with all types of cancer.
2. In the introduction, when the authors refer to data from reference 40, it must be in the hepatocarcinoma cell line. Authors need to denote very carefully when it is in a model of human cell lines and when it is in tumoral samples of patients. Any result obtained from a model of the cell line can not be generalized to cancer in a general sense. Have to be very careful.
3. In line 123, The authors describe that:
"TAT- IF1-O.1 target the IF1sequence overlapping a small part of coiled-coil domain which allows the formation of IF1 dimers", why authors say in this sentence:
"confirming the specificity of IF1-O.1 for targeting the IF1-OSCP interaction".
Authors have mentioned that the "IF1-0.2: target the N-terminal region of the OSCP subunit which was previously shown to be involved in the interaction with IF1."
4. Line 126, revise:
"human OSCP subunit of ATP synthase obtained as in 24.
5. In discussion:
"Importantly, the use of mitochondria-targeting peptides directed to the IF1-OSCP interaction or to the C-terminal region of the OSCP subunit mimic or counteract respectively the anti-apoptotic effects of the high levels of IF1 in cancer cells by modulating the PTP opening.
The authors did not prove this effect. Revise.
Author Response
Reviewer 1
Comments and Suggestions for Authors
The manuscript entitled: "Peptides targeting the IF1-ATP synthase complex modulate the permeability transition pore in cancer cells" was revised. The manuscript was presented in order in all sections. The design was optimal and adequate to answer the permeability transition pore question. The manuscript has minor observations:
- The title must be more adequate. According to their results, it must be "in HeLa cells" or "tumoral cell line HeLa" because authors can not associate it with all types of cancer.
We would like to thank the Reviewer 1 for this comment. The title has been changed into “Peptides targeting the IF1-ATP synthase complex modulate the permeability transition pore in cancer HeLa cells”
- In the introduction, when the authors refer to data from reference 40, it must be in the hepatocarcinoma cell line. Authors need to denote very carefully when it is in a model of human cell lines and when it is in tumoral samples of patients. Any result obtained from a model of the cell line can not be generalized to cancer in a general sense. Have to be very careful.
Thank you for the suggestion, the revised text has been improved accordingly, as it follows on lines 61-63:
“In hepatocellular carcinoma cells IF1 was shown to promote cell proliferation and colony formation in vitro, by decreasing the expression of E-cadherin and increasing STAT3 level 40.”
- In line 123, The authors describe that:
"TAT- IF1-O.1 target the IF1sequence overlapping a small part of coiled-coil domain which allows the formation of IF1 dimers", why authors say in this sentence:
"confirming the specificity of IF1-O.1 for targeting the IF1-OSCP interaction".
Authors have mentioned that the "IF1-0.2: target the N-terminal region of the OSCP subunit which was previously shown to be involved in the interaction with IF1."
We thank this Reviewer for this insightful comment, the text has been improved as follows on lines 121-124:
“This effect on the IF1 binding to the enzyme was not observed when the IF1-O.2 or IF1-O.3 peptides were used in incubation (Figure 2A), confirming the specificity of IF1-O.1 for targeting the IF1 inhibitor and causing its displacement from the OSCP subunit under ATP synthesis condition.”
- Line 126, revise:
"human OSCP subunit of ATP synthase obtained as in 24.
We thank the Reviewer for this comment, the text has been improved as follows on lines 128-130:
“… the recombinant N-terminal domain (R6-G114) of the human OSCP subunit of ATP synthase, which was previously found to be involved in the interaction with IF1 24.”
- In discussion:
"Importantly, the use of mitochondria-targeting peptides directed to the IF1-OSCP interaction or to the C-terminal region of the OSCP subunit mimic or counteract respectively the anti-apoptotic effects of the high levels of IF1 in cancer cells by modulating the PTP opening.
The authors did not prove this effect. Revise.
We thank this Reviewer for the suggestion, the text has been improved as follows on lines 294-295:
“… mimic or counteract, respectively, the effects of the high levels of IF1 in HeLa cancer cells by modulating the PTP opening.”
Reviewer 2 Report
Comments and Suggestions for Authors
The paper describes the interaction of 3 membrane permeable peptides with the IF1-ATP synthase complex. I have the following comments:
1. Why IF1-O.2 has not been studied in the immunoprecipitation assay?
2. Is there any pertubation of the NMR signal by IF1-O.1?
3. In Fig. 5 it would be more informative if absolute values of Ca2+ per μg of protein that cause PTP opening would be shown.
4. I would strongly recommend to shift the important Fig. S2 to the main paper. A statistical evaluation of the degree of IF1 KD in the different experiments would be also desirable.
Comments on the Quality of English LanguageNone.
Author Response
Reviewer 2
Comments and Suggestions for Authors
The paper describes the interaction of 3 membrane permeable peptides with the IF1-ATP synthase complex. I have the following comments:
- Why IF1-O.2 has not been studied in the immunoprecipitation assay?
We would like to thank this Reviewer for this insightful comment, in the revised manuscript the immunoprecipitation in the presence of IF1-O.2 has been shown and the results has been described on lines 121-124 and in the new Figure 2A:
“This effect on the IF1 binding to the enzyme was not observed when the IF1-O.2 or IF1-O.3 peptides were used in incubation (Figure 2A), confirming the specificity of IF1-O.1 for targeting the IF1 inhibitor and causing its displacement from the OSCP subunit under ATP synthesis condition.”
- Is there any pertubation of the NMR signal by IF1-O.1?
We would like to thank the Reviewer 2 for this suggestion. We have performed the NMR analysis in the presence of IF1-O.1 on the OSCP recombinant N-terminus (figure in the pdf of the point-by-point response to Reviewers), but this experiment did not show significant chemical shift perturbation at the same molar excess (10-fold concentration) of peptide IF1-O.2, which was used for the NMR experiment in Figure 2B. This is not surprising because the peptide IF1-O.1 was not designed to interact with OSCP-NT, however, given the remarkable sequence similarity with peptide IF1-O.2 a weak interaction with OSCP-NT is possible. Therefore, we cannot exclude that higher concentrations, that are not testable in this kind of NMR measurements for technical limitations (i.e. peptide oligomerization), might cause shift perturbations of the OSCP fragment. Accordingly, a higher concentration of peptide IF1-O.1 than IF1-O.2 was required in the CRC experiments in order to see a significant increase in PTP sensitivity to Ca2+ in IF1 KD HeLa cells (new figures 6B and 6C). Since the effect of this peptide IF1-O.1 on the OSCP N-terminus is not fully characterized we prefer not to show the NMR spectra in the revised paper.
- In Fig. 5 it would be more informative if absolute values of Ca2+ per μg of protein that cause PTP opening would be shown.
We thank the Reviewer for this comment. To address this issue, the nmols of Ca2+/µg of protein were shown for control and IF1 KD HeLa cells in the new Figure 6 of the revised manuscript. However, we preferred to show the effects of peptides after normalization (as % of the untreated cells) because we think that this representation more easily shows to the reader the effects of peptides independently of the differences between the two genotypes.
- I would strongly recommend to shift the important Fig. S2 to the main paper. A statistical evaluation of the degree of IF1 KD in the different experiments would be also desirable.
We strongly thank the Reviewer for this suggestion. Data are shown in the new Figures 4 and 6 of the revised manuscript. We think this shift greatly improved our manuscript.
Reviewer 3 Report
Comments and Suggestions for Authors
I review the article by Grandi et al., entitled “Peptides targeting the IF1-ATP synthase complex modulate the permeability transition pore in cancer cells.” This paper makes good understanding the molecular mechanisms of the interaction between IF1 and the mitochondrial ATP synthase complex, especially by identifying the binding of specific peptides to the OSCP subunit using NMR spectroscopy analysis. Furthermore, the manuscript validates through a series of experiments the role of these peptides in regulating mitochondrial metabolism and the opening of the PTP in cancer cells, providing important biological insights for the development of new anti-cancer strategies. However, I would like the authors to consider the following suggestions for revisions or additions:
1. the authors conduct biological experiments using at least two or more cancer cell lines to made conclusions. This would help validate the universality and reproducibility of the results.
2. When performing OCR measurements, the authors better quantify the cells effectively to ensure that the number of cells in the KD group and the control group is consistent. Given that the initial OCR values of the KD and CTRL groups in the article are very quite similar at 600 pmol/min, this is particularly important to eliminate biases in experimental design or operation.
3. The authors need to supplement the long-term effects of peptides on the growth and survival of cancer cells. At the same time, seeing the results of cell viability tests for the control group and IF1-KD group would be very valuable.
4. The authors could do experiments that measuring mitochondrial membrane potential and the expression of respiratory chain proteins to strengthen the evidence for the impact on mitochondrial bioenergetics. Additionally, this could indirectly reveal the impact of Ca2+ on MMP, supporting the PTP related results.
minor comments:
- The authors could include in the discussion how these findings could be translated into concrete anticancer treatments, including how to address potential drug resistance issues.
- The authors could use TEM to observe mitochondrial morphology and structure. Alternatively, the authors could also use MitoTracker and Ca2+ probes for live-cell localization to assess the impact of Ca2+ load on mitochondrial network and structure.
- The authors can supplement experiments on the expression of mitochondrial respiratory chain proteins, both on protein level or mRNA level are fine, to enhance the evidence for the impact on mitochondrial bioenergetics.
Comments on the Quality of English LanguageThe English language quality of the manuscript by Grandi et al., titled “Peptides targeting the IF1-ATP synthase complex modulate the permeability transition pore in cancer cells,” is generally good, allowing for clear understanding of the research conducted and the findings presented.
Author Response
Reviewer 3
Comments and Suggestions for Authors
I review the article by Grandi et al., entitled “Peptides targeting the IF1-ATP synthase complex modulate the permeability transition pore in cancer cells.” This paper makes good understanding the molecular mechanisms of the interaction between IF1 and the mitochondrial ATP synthase complex, especially by identifying the binding of specific peptides to the OSCP subunit using NMR spectroscopy analysis. Furthermore, the manuscript validates through a series of experiments the role of these peptides in regulating mitochondrial metabolism and the opening of the PTP in cancer cells, providing important biological insights for the development of new anti-cancer strategies. However, I would like the authors to consider the following suggestions for revisions or additions:
- the authors conduct biological experiments using at least two or more cancer cell lines to made conclusions. This would help validate the universality and reproducibility of the results.
We thank Reviewer 3 for this suggestion, we are aware that one cell line is not enough to draw general conclusions on the studied peptides. In order to address this point, and point 1 of Reviewer 1, we elucidated in the revised text that our study was focused only in HeLa cells. The title has been changed into “Peptides targeting the IF1-ATP synthase complex modulate the permeability transition pore in cancer HeLa cells”, as also requested by Reviewer 1. Thanks to this suggestion we will explore the possibility of testing these peptides in a larger panel of cancer cell types in the next studies.
- When performing OCR measurements, the authors better quantify the cells effectively to ensure that the number of cells in the KD group and the control group is consistent. Given that the initial OCR values of the KD and CTRL groups in the article are very quite similar at 600 pmol/min, this is particularly important to eliminate biases in experimental design or operation.
We thank Reviewer 3 for this request, we have added a new Figure S2 in the revised manuscript, showing OCR quantification/ µg of protein (quantified in each well, Fig. S2 A and B). These data are not in contrast with the CTR (plko- empty vector) and IF1 KD cell OCRs shown in Figure 4 of the previous version of the manuscript. For this reason, we decided to keep the cell respiration traces as Figure 5 in the main set of figures of the revised paper. The absence of discrepancy between the OCR values of 50000 CTR and IF1 KD cells per well (in new Figure 5) and the normalized OCR values (in new Figure S2A and B) is in line with the comparable proliferation curves of CTR and IF1 KD HeLa cells, which are shown in the new Figure S2C.
- The authors need to supplement the long-term effects of peptides on the growth and survival of cancer cells. At the same time, seeing the results of cell viability tests for the control group and IF1-KD group would be very valuable.
We thank Reviewer 3 for this suggestion. A growth curve with or without a 48-hour treatment with peptide IF1-O.3, which causes PTP opening and theoretically might exert cell toxicity, is shown in Fig. S2C of the revised manuscript, revealing no significant toxicity in control and IF1 KD HeLa cells.
- The authors could do experiments that measuring mitochondrial membrane potential and the expression of respiratory chain proteins to strengthen the evidence for the impact on mitochondrial bioenergetics. Additionally, this could indirectly reveal the impact of Ca2+ on MMP, supporting the PTP related results.
Thanks to this suggestion, in the new Figure 4 of the revised manuscript the OXPHOS quantification in CTR and IF1 KD HeLa cells and their normalization are shown. The treatment with the three peptides does not affect the OXPHOS levels, nor the membrane potential (as shown by the levels of basal and FCCP-stimulated OCRs in the new Figure 5, in the presence of the three peptides). However, to clarify these aspects in accordance to this Reviewer comment the text has been improved on lines 205-208:
“Moreover, the peptide-treatment did not affect the mitochondrial membrane potential of HeLa cells, nor the OXPHOS maximal activity (as shown by the comparison between the basal and FCCP-stimulated respiration, upon peptide treatment, Figure 5).”
minor comments:
- The authors could include in the discussion how these findings could be translated into concrete anticancer treatments, including how to address potential drug resistance issues.
We thank this Reviewer for the comment. The manuscript has been improved on lines 352-356:
“This latter result indicates the protein fragment which can be targeted in future studies to develop anti-cancer compounds. In order to better define the therapeutic potential of the synthetic peptides shown here, that were useful to address the molecular mechanism of the IF1-OSCP binding in HeLa cells, their effects should be tested on other cell types and in in vivo cancer models.”
- The authors could use TEM to observe mitochondrial morphology and structure. Alternatively, the authors could also use MitoTracker and Ca2+ probes for live-cell localization to assess the impact of Ca2+ load on mitochondrial network and structure.
We would like to thank the Reviewer 3 for this suggestion. We will keep this in mind when exploring the long-term effects of these peptides in the future studies.
- The authors can supplement experiments on the expression of mitochondrial respiratory chain proteins, both on protein level or mRNA level are fine, to enhance the evidence for the impact on mitochondrial bioenergetics.
We would like to thank the Reviewer 3 for this suggestion. We have shown in the new Figure 4 of the revised manuscript the OXPHOS protein quantification in CTR and IF1 KD HeLa cells and their normalization. The short-term treatment with the three peptides does not affect the OXPHOS levels, therefore as stated in one of the previous points for this Reviewer the text has been improved on lines 205-208:
“Moreover, the peptide-treatment did not affect the mitochondrial membrane potential of HeLa cells, nor the OXPHOS maximal activity (as shown by the comparison between the basal and FCCP-stimulated respiration, upon peptide treatment, Figure 5).”
Round 2
Reviewer 2 Report
Comments and Suggestions for Authors
All of my comments have been addressed accordingly.
Author Response
We thank the Reviewers for their suggestions.